# Dominance of the scleractinian coral *Alveopora japonica* in the barren subtidal hard bottom of high-latitude Jeju Island off the south coast of Korea assessed by high-resolution underwater images

**Kyeong-Tae Lee**[1,2], **Hye-Mi Lee**[1], **Thatchaneshkanth Subramaniam**[1], **Hyun-Sung Yang**[2], **Sang Rul Park**[1], **Chang-Keun Kang**[3], **Shashank Keshavmurthy**[4]*, **Kwang-Sik Choi**[1]*

1 Department of Marine Life Science (BC 21 PLUS) and Marine Science Institute, Jeju National University, Jeju, Republic of Korea, 2 Jeju Marine Research Center, Korea Institute of Ocean Science & Technology (KIOST), Jeju, Korea, 3 School of Earth Sciences and Environmental Engineering, Gwangju Institute of Science and Technology, Gwangju, Republic of Korea, 4 Biodiversity Research Centre, Academia Sinica, Nangang, Taipei, Taiwan

* skchoi@jejunu.ac.kr (KSC); coralresearchtaiwan@gmail.com (SK)

## Abstract

Coastal benthic communities in temperate regions have been influenced by climate change, including increasing sea-surface temperature. Nevertheless, scleractinian coral *Alveopora japonica* Eguchi, 1968, is thriving in shallow subtidal hard bottoms around Jeju Island, off the southern coast of Korea. The presence of this corals has negatively impacted subtidal kelp populations in Jeju Island. However, there is no study to document how the presence or absence of this coral relates to other benthic communities. This study investigated the benthos in three shallow subtidal sites (Shinheung (SH), Bukchon (BC), and Seongsan (SS)) in northern Jeju using underwater photography. Macro-benthic organisms appearing on a 1 × 20 m line transect installed at depths of 5, 10, and 15 m at each site were analyzed. Results showed that of the three sites investigated, *A. japonica* colonies were most abundant at BC, accounting for 45.9% and 72.8% of the total transect area at 10 m and 15 m, respectively. At SS, *A. japonica* occupied 15.3% of the total area at 15 m and less than 1% at 5 m and 10 m. The same at SH accounted for 10% of the total area at 5 m, and less than 1% at 10 m and 15 m. Dead and bleached colonies accounted for 1.2–11.5% and 1.8–5.7%, respectively, at 5, 10, and 15 m at three sites. At SS, canopy-forming brown algae *Ecklonia cava* and *Sargassum* spp. accounted for 20.2 and 24.3% of the total transect area, respectively, at 5 m depth. In contrast, the percent cover of *E. cava* and *Sargassum* spp. at SH and BC ranged from 0.1 to 1.8%, respectively. Moreover, non-geniculate coralline algae dominated the subtidal substrate at SH, ranging between 60.2 and 69% at 15 and 10 m. The low cover of *A. japonica* in SS (at 5 m) coincided with a high percent cover of canopy-forming brown algae. However, canopy-forming brown algae were rare at all depths at SH and BC and were dominated instead by coralline algae and the scleractinian corals. This study, by utilizing a non-destructive method, provides a baseline qualitative and quantitative information for

**Data Availability Statement:** All relevant data are within the paper and its Supporting Information files.

**Funding:** This study was supported by the Basic Science Research Program through the National Research Foundation of Korea (NRF), which is funded by the Ministry of Education, [2019R1A6A1A03033553] and by "Responses of Species-Populations to Climate Change Scenario" program of Korea institute of Marine Science & Technology Promotion (KIMST), funded by the Ministry of Oceans and Fisheries, [KIMST-20220559] in the form of grants to K-SC; by the Korea Institute of Ocean Science & Technology in the form of a grant to H-SY [PEA0016]; and by a postdoctoral fellowship at Academia Sinica awarded to SK. The funders had no role in study design, data collection and analysis, decision to publish, or preparation of the manuscript.

**Competing interests:** The authors have declared that no competing interests exist.

understanding the site and depth-dependent distribution of *A. japonica* and algal populations, which is important to understand climate change related changes in benthic communities in Jeju and elsewhere.

## Introduction

The consequences of climate change-related changes, potentially deleterious, needs to be monitored to understand, protect or minimize the impact on marine ecosystems [see 1]. The structure and composition of the benthic community, escalated by natural and anthropogenic activities, may be rearranged by climate change. These changes may cause an evident decline in the population of the benthic species that fail to adapt and/or acclimatize to changing climate, resulting in new interactions with species colonizing the ecosystems. In particular, the association might be altered between benthic organisms when the keystone species of that particular marine ecosystem disappear [2–4] and introducing species from tropical locations. This idea has already been explored in several studies in recent years under the premise of the "tropicalization" of high-latitude communities [4, 5]. In benthic ecosystems, space is a limiting factor for sessile flora and fauna, affecting recruitment, growth rate, and distance from other populations [6, 7].

Several studies have been carried out on the benthic species composition in the waters of Jeju Island [8–10], mainly concerning the dominant coral *Alveopora japonica* [11–14]. This scleractinian coral occurs commonly in non- reefal temperate waters and is widely distributed from Taiwan to Japan, including Jeju Island, off the south coast of Korea [14, 15]. In Jeju Island, this species dominates mainly in the northern part of the Island (between 5–20 m in depth) and associates and shares space with mollusks, algae, and soft corals [12, 16, 17]. According to Vieira et al. [2016], *A. japonica* has a short life span [12–13 years] and fast turnover rates with a growth rate of 4.8mm year$^{-1}$. Such growth and turnover rates could be the reason for the high density of *A. japonica* (ranging from 58 to 155 colonies m$^{-2}$) at several locations in Jeju [12, 17]. Furthermore, in Jeju, this species recruits and re-populates abundantly with an estimated recruitment rate of 7.8 colonies m$^2$ yr$^{-1}$ via sexual reproduction [13, 14]. A detailed study on the assemblage of molluscans association with *A. japonica* collected from several locations in Jeju has been reported by Noseworthy et al. [2016]. While increased abundances of *A. japonica* could be a factor for the decline of canopy-forming brown algae in Jeju, crustose coralline algae have considerable coverage in some locations and have replaced large macroalgae in density [11, 12, 18]. Anecdotal evidence has reported that until 1980, brown macroalgae (*Sargassum* sp., *Laminaria* sp., and *Ecklonia* sp.) were dominant in the shallow subtidal rocky bottom, playing a vital role, ecologically and economically, in Jeju Island [12, 19]. The brown macroalgal beds formed on the shallow subtidal areas were utilized as breeding grounds for finfish and shellfish and were crucial for the juveniles [19–21]. However, in the past 2–3 decades, the macroalgal communities in high latitude, including Jeju, have been decimated due to the effect of temperature and other factors (feeding by herbivores such as sea urchins, invasion, and herbivory by tropical fish; see [22, 23]–Kochi, Japan; [24]–Jeju-South Korea), resulting in barren grounds, known as 'getnoguem' in Korean and 'isoyake' in Japan [25], with coralline algae and little else covering the rocks. This has led to coral takeover in many locations (see [13, 16]–Jeju and [26, 27]–Kochi), with increased recruitment and abundance of several coral species. While several studies have reported biodiversity of the benthic organisms in Jeju Island, no studies examined the ecological association of *A. japonica* with

macroalgae and other benthos at a spatial scale (horizontally–at different locations and vertically- at different depths).

Studies related to benthic community dynamics have utilized various techniques, including line intercept transect, point intercept transect, photo-quadrats, and video transect [see 28–37]. A higher degree of precision has become possible with recent advances in photography and image classification tools in various quantitative methods [38]. Underwater imaging has improved the quality of the surveys due to the availability of high-resolution imagery [33, 35, 38–40]. Moreover, if the area survey is not very large, the number of photos obtained is still manageable, and their analysis can be performed in a relatively short time. Photographic data enables quantitative monitoring of physical (e.g., structural complexity: slope, fractal dimension, surface complexity) and biological features (e.g., the cover of benthic communities, colonies size, and abundance) of ecosystems over time [33, 35, 38–40].

The main purpose of the present study was to analyze benthic community structure and dynamics at three sites in Jeju Island by utilizing high-resolution photography (a non-destructive method). We discuss the diversity of benthos, the abundance and relationship of *A. japonica* with the benthos in Jeju Island, and the possible role of different benthic organisms at three sites and depths.

## Materials and methods

### Study site

The study was conducted at three sites, Shinheung (SH); Bukchon (BC); and Seongsan (SS), along the northeast and east coast of Jeju Island, off the south coast of Korea (Fig 1). Jeju Island is located at off the southwestern tip of the Korean peninsula, strongly influenced by the warm Tsushima Current [12, 37]. As a result, Jeju Island is characterized as a warm, humid climate with four distinct seasons, and the average annual seawater temperature ranges from 13.7˚C to 25˚C [14].

### Underwater images collection

Twenty high-resolution images were captured (1m × 1m), in January 2014, continuously over a 20 × 1 m line transect (one transect at each depth at each location) at 1 m intervals during every 30 minutes dives using SCUBA with a digital underwater camera at depths of 5, 10, and 15 m, respectively (Fig 2). The diver used a Nikon D90 digital camera (a maximum resolution of 4288 × 2848 pixels; using Patima housing) at a vertical position away from the rocky subtidal substrate to take photographs. The camera housing was equipped with two strobes to provide additional light. The total area examined was 180 m$^2$ for this study.

### Image analysis

The image analysis was performed using photoQuad ® software that operates in a layer-based environment, following multiple computations on the same source photograph [41]. Line transects were marked at every meter with a plastic plate showing the serial number from 0 to 20 on a 20 m transect laid on the benthos. The photos captured at regular intervals underwater on the transect were used to determine the scale from each photo showing the serial number of the one-meter interval. The underwater digital images were outlined in 1 m$^2$ virtual photo quadrats and converted to actual distance from the pixel size using the software tools. Following this, the abundance of *A. japonica* occurring on the transects was estimated using freehand regions and the random point counts method (see S1 File for how the images were analyzed)

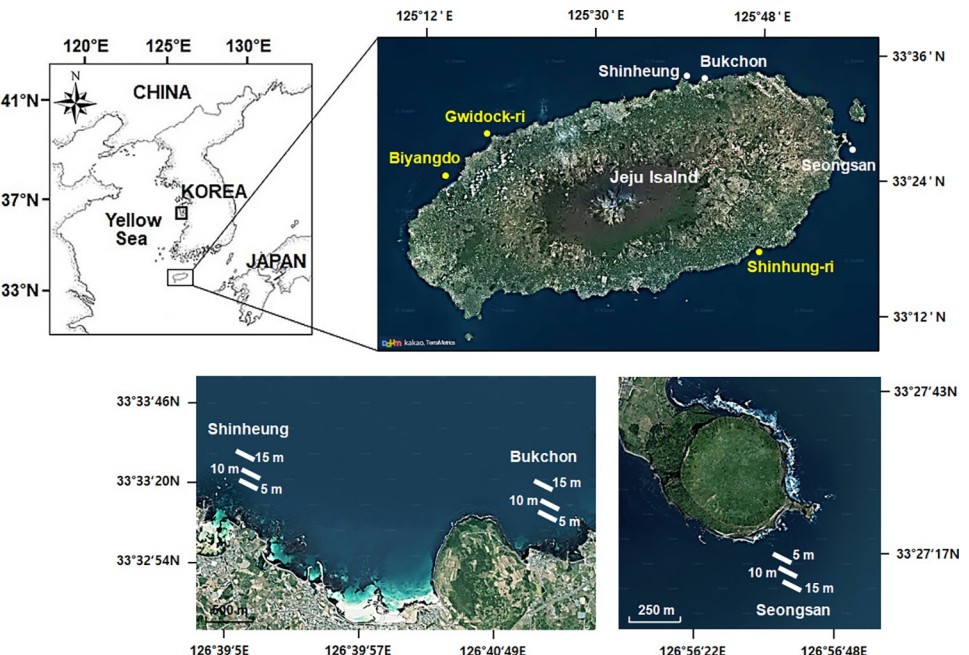

**Fig 1. Map showing the sampling locations of 20-m line transects installed at depths of 5 m, 10 m, and 15 m at three survey sites SH, BC, and SS in Jeju Island, Korea.** SH = Shinheung, BC = Bukchon, SS = Seongsan, GW = Gwidock-ri, BD = Biyangdo, GN = Geumneung, SH-ri = Shinhung-ri. The locations depicted in white are from this study, and those in yellow are from previously published studies. (Map Source: https://map.kakao.com).

The freehand regions method was used to count the number of *A. japonica* colonies in the transects (Fig 3, lines in blue). Bleached and dead colonies of *A. japonica* could be identified from the photo quadrats. Accordingly, we categorized the coral colonies into (1) live colonies with long extended polyps, (2) bleached colonies with white coral tissue, and (3) dead colonies

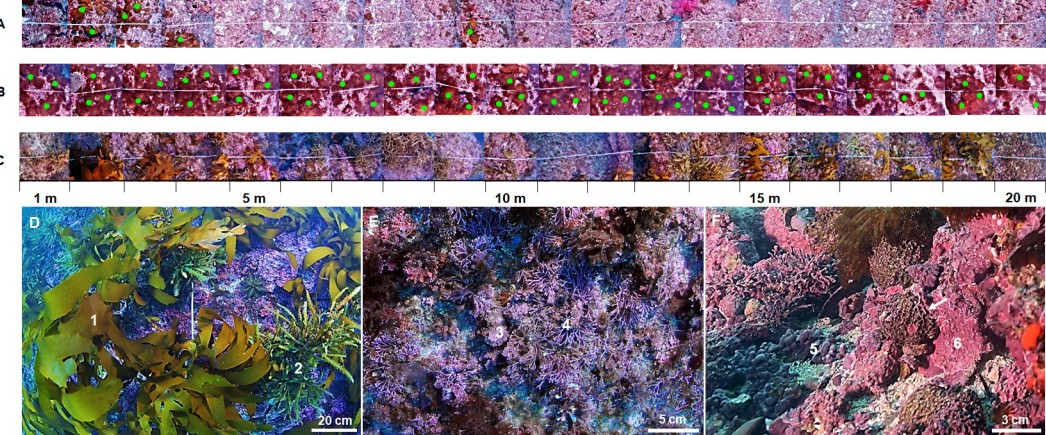

**Fig 2.** 20-m line transects installed at (A) a depth of 5 m of SH (B) a depth of 15 m of BC (C) a depth of 5 m of SS in Jeju Island, Korea. Photographs were captured continuously at 1 m interval (20 photographs per transect) using an underwater digital camera (a full resolution of 4288 × 2848) with a 1 m standoff of the substrate over the transect. Green points indicate *A. japonica* appeared at 20-m line transects. (D) the prolific growth of large canopy-forming brown algae 1) *Ecklonia* sp., 2) *Sargassum* sp.; (E) geniculate growth form 3) *Corallina* sp., 4) *Amphiroa* sp.; (F) non-geniculate growth forms 5) rhodolith, 6) crustose (arrow) overgrowing the dead colony (*A. japonica*).

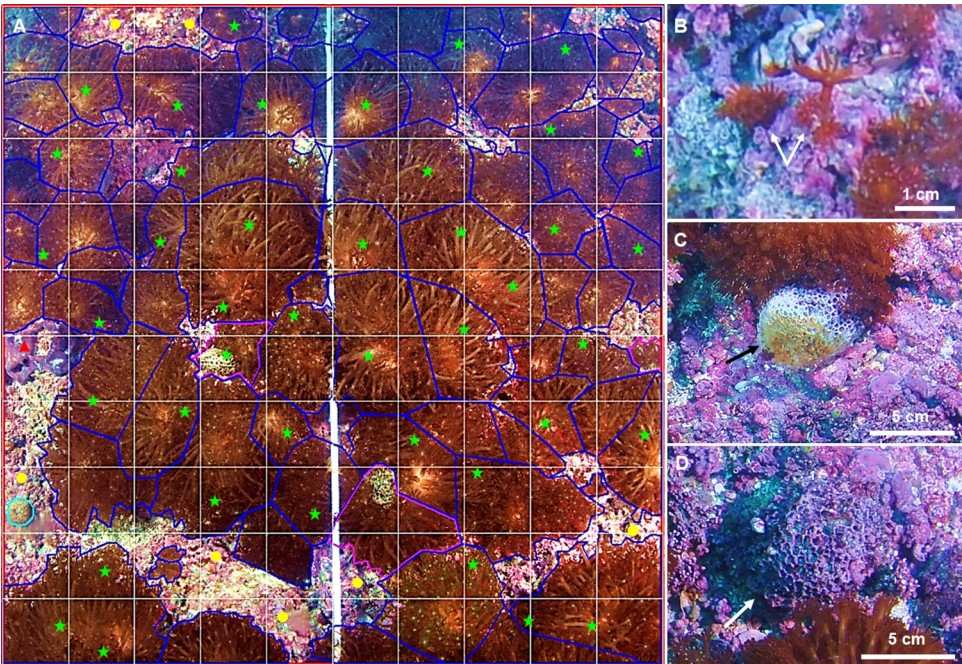

**Fig 3.** (A) 1m² photo quadrat taken from a depth of 15 m of BC using an underwater camera and image analysis methods of freehand regions and random point counts using photoQuad software. The freehand analysis involved the manual drawing of outline around *A. japonica* colonies. Blue lines indicate live colonies, pink lines indicate bleached colonies, and skylines indicate dead colonies. The random point counts were performed on 50 randomly allocated points to benthic organisms. Green points indicate *A. japonica*, yellow points indicate non-geniculate corallines, and redpoint indicate *Lissoclinum* sp. appeared on a 1m² photo quadrat. (B) a close-up picture of small colonies (arrow) of *A. japonica* identified from an underwater image. (C) Bleached colony (arrow) with coral tissue lost its color and whiting colonies. (D) the dead colony (arrow) with all coral tissue disappeared.

with only skeleton remaining, following [42, 43]. The minimum size of *A. japonica* identified as individual colonies on a PC monitor screen was 1 cm in diameter.

The random point counts method was used to estimate the percent cover of benthic organisms, including *A. japonica* in the transects (Fig 3, green and yellow stars). We selected 50 randomly assigned points for each outlined 1 m² virtual photo quadrat and identified the underlying benthic organisms according to visual information such as color, texture, shape, and size. The percent cover of benthic organisms was estimated from 1,000 random points per transect. The points were excluded from the analysis if the allocated point was unclear (i.e., shadow or blurry) or objects located in the sand. Coralline algae assemblages were grouped based on their growth forms into non-geniculate growth forms and geniculate growth forms, as described in [44]. The benthic organisms were assigned into one of the six categories, 1) hard coral (exclusively *A. japonica*), 2) other sessile invertebrates (i.e., soft corals, sponges, mollusks, bryozoans, echinoderms, ascidians, anemones, and annelids), 3) canopy forming brown algae (includes kelp: *Ecklonia cava* and fucoids: *Sargassum* spp.), 4) geniculate corallines algae, 5) non-geniculate corallines including both crustose and rhodolith forms, and 6) other fleshy macroalgae, such as chlorophyte, phaeophyte, and rhodophyte excluding canopy-forming brown algae and coralline algae (genticulate and non-genticulate corralines).

## Data analysis

Cluster analysis based on the Bray-Curtis coefficient was performed to evaluate similarity by the percent cover of each benthic organism in the transects. We also ran the similarity profile

routine (*SIMPROF, [45]*) and the analysis of similarity (ANOSIM, [46]), and the similarity percentages (SIMPER, [46]) to examine the contribution of individual species or functional groups (i.e., non-geniculate corallines, geniculate corallines) to the similarities calculated and used in the cluster analysis. All statistical analyses were performed using PRIMER software (PRIMER-E Ltd).

## Results

### Analysis of benthic composition

Photographs from a total of 9 transects from three depths at three sites Shinheung (SH, Bukchon (BC), and Seongsan (SS), were analyzed to understand the benthic composition. The benthic organisms of various sizes were recorded, from relatively small-sized species (i.e., *Herdmania* sp., *Colpomenia* sp.) to large-sized species (i.e., *Dendronephthya* sp., *Ecklonia cava*). The benthic organisms and their percent coverage in the transects are summarized in S1 Table and S2–S4 Figs in S2 File. *Alveopora japonica* colonies were present in all three sites and were most abundant at BC (Table 1, Fig 4). At SS, macro-algae were the most abundant and diverse, mainly the canopy-forming brown algae *Ecklonia cava* and *Sargassum spp.* occurred at 5 m depth. In contrast, non-geniculate corallines covered the rocky substrate widely at BC and SH (Fig 4).

Fig 5 shows the average cover of the benthic community estimated by analyzing the underwater photographs. At BC, *A. japonica* accounted for 45.9 (10 m) and 72.8% (15 m) of the transect cover. At SS 15 m depth, *A. japonica* occupied 15.3% on the transect and less than 1% at depths of 5 and 10 m. At SH, *A. japonica* accounted for 10.0% of the total area at 5 m depth, whereas the coverage was limited to less than 1% at depths of 10 and 15 m. However, at 5 m depth of SS, the coverage of the canopy-forming brown algae was 44.5% on the transect, and geniculate corallines occupied 36.9% on the transect. At SS 10 m depth, major categories contributing to the coverage were other fleshy macroalgae (47.2%) and geniculate corallines (25.3%). In contrast, canopy-forming brown algae ranged from 0 to 1.8%, and other fleshy macroalgae ranged from 0.1 to 4.9%, respectively, at SH and BC. Non-geniculate corallines

**Table 1. Summary of the percentage cover and density of *A. japonica* reported from sites in Jeju Island, Korea (see Fig 1 legend for the explanation of site names).**

| Site name | Survey area (m²) | Depth (m) | Percent cover (%) | Density (colonies/m²) | Site | Reference |
|---|---|---|---|---|---|---|
| SH | 20 | 5 | 10.0 ± 3.1 | 29.0* | 33°33′22″N, 126°39′14″E | Present study |
| | | 10 | 0.8 ± 0.4 | 0.8* | | |
| | | 15 | 0.2 ± 0.1 | 0.1* | | |
| BC | 20 | 5 | 0.4 ± 0.3 | 0.2* | 33°33′12″N, 126°41′11″E | Present study |
| | | 10 | 45.9 ± 3.6 | 58.4* | | |
| | | 15 | 72.8 ± 2.4 | 65.7* | | |
| SS | 20 | 5 | 0.1 ± 0.1 | 0.1* | 33°27′14″N, 126°56′39″E | Present study |
| | | 10 | 0 | 0* | | |
| | | 15 | 15.3 ± 2.4 | 40.4* | | |
| BD | 15.75 | 15 | 67 ± 4 | Na | 33°24′36″N, 126°13′12″E | Denis et al. 2015 |
| | 1 | 15 | na | 7590 | 33°24′36″N, 126°13′12″E | Denis et al. 2015 |
| GD | 10 | 10 | 75 | 87.8 | 33°26′56″N, 126°17′33″E | Vieira et al. 2016 |
| SH-ri | 10 | 10 | 75 | 154.7 | 33°17′44″N, 126°46′13″E | Vieira et al. 2016 |
| GN | 10 | 10 | na | 57.9 | 33°23′42″N, 126°13′08″E | Noseworthy et al. 2016 |

*Na*, not available

*, whole colonies (live colony + bleached colony + dead colony

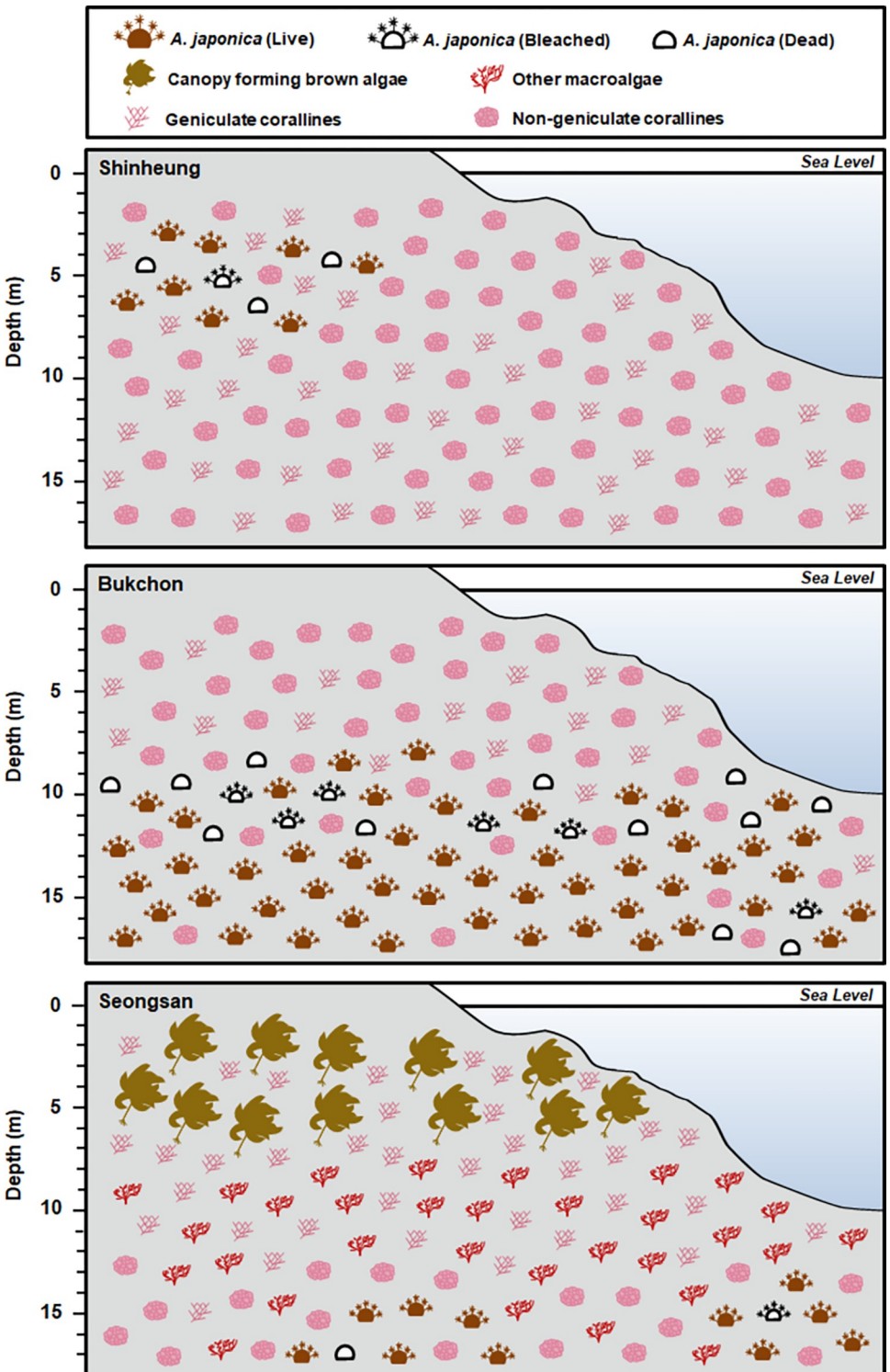

**Fig 4. Illustration of major benthic community and different conditions: Live, bleached, and dead colonies of *A. japonica* occurring at different depths in 3 survey sites SH, BC, SS in Jeju Island, Korea.**

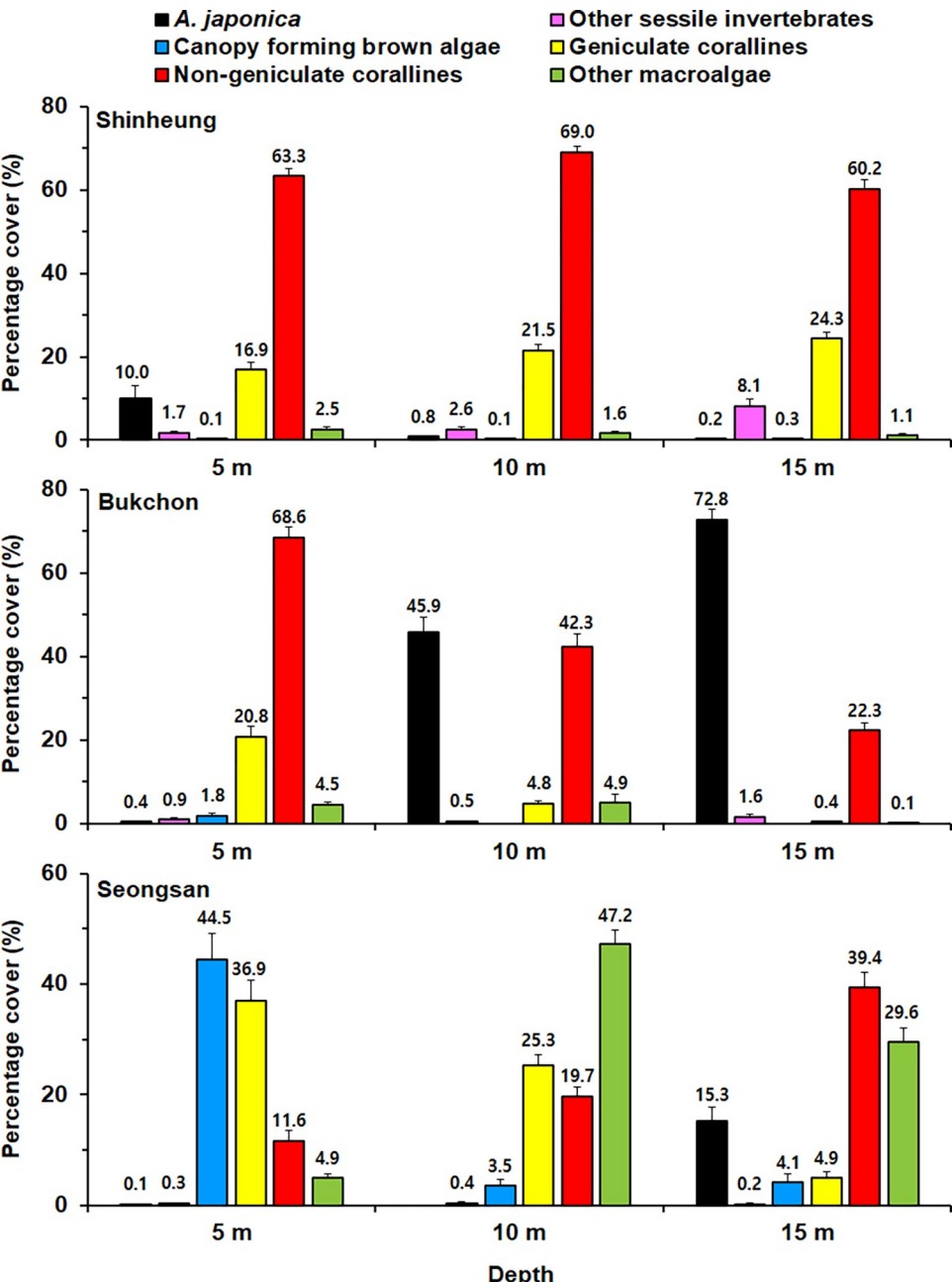

**Fig 5. Average percent cover of benthic community estimated on 20-m line transects installed at depths of 3 survey sites SH, BC, SS.** The percent cover is followed by the standard error.

dominated the rocky substrate, ranging from 60.2 to 69.0% of the total area at all depths of SH and BC 5 m. At SH, other sessile invertebrates ranged from 1.7 to 8.1%, and their abundance was higher than BC and SS.

## *Alveopora japonica* population dynamics

The underwater photography revealed considerable differences in the percent coverage and density of *A. japonica* within and between sites (Table 1, Fig 6). The density of *A. japonica* in

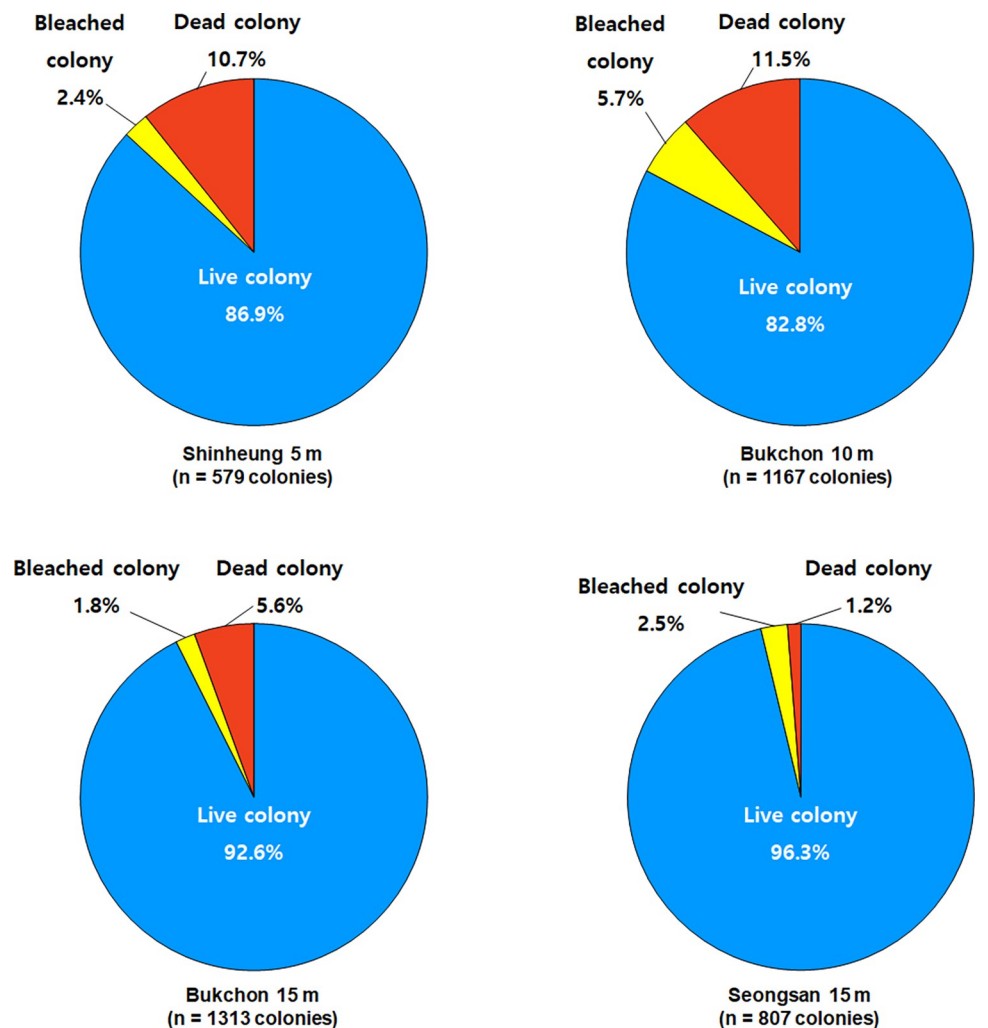

**Fig 6. The prevalence (%) of different conditions (live, bleached, and dead colony) of *A. japonica* colonies observed on 20-m line transects installed at depths of 3 survey sites SH, BC, SS.** The total number (n) of colonies (live + bleached + dead) analysed at each site is shown below the site name of each pie chart.

BC was 58.4 and 65.7 colonies m$^{-2}$ at 10 m and 15 m depths. On the contrary, the coral density was 29.0 and 40.4 colonies m$^{-2}$ at 5 m and 15 m depths at SH and SS, respectively. No *A. japonica* colonies were recorded from the line transect installed at 10 m depth at SS. The range of bleached and dead colonies of *A. japonica* was between 1.8–5.7% and 1.2–11.5% respectively, in the study sites, with the highest percentage (96.3%) of live colonies recorded at 15 m depth at SS (Fig 5). Even though the highest coral coverage and density were seen at BC, surprisingly, the highest percentage of dead colonies (11.5%) and bleached colonies (5.7%) were reported at 10 m depth of BC as well (Fig 6)

## Cluster analysis

Cluster analysis based on the percent cover of benthic organisms estimated from the transects showed two distinct groups corresponding to the northeast coast (Group 1) and east coast (Group 2) of Jeju Island (Fig 7). The cluster analysis also suggested that Group 1 could be sub-divided into two subgroups A and B. ANOSIM indicated significant differences between the

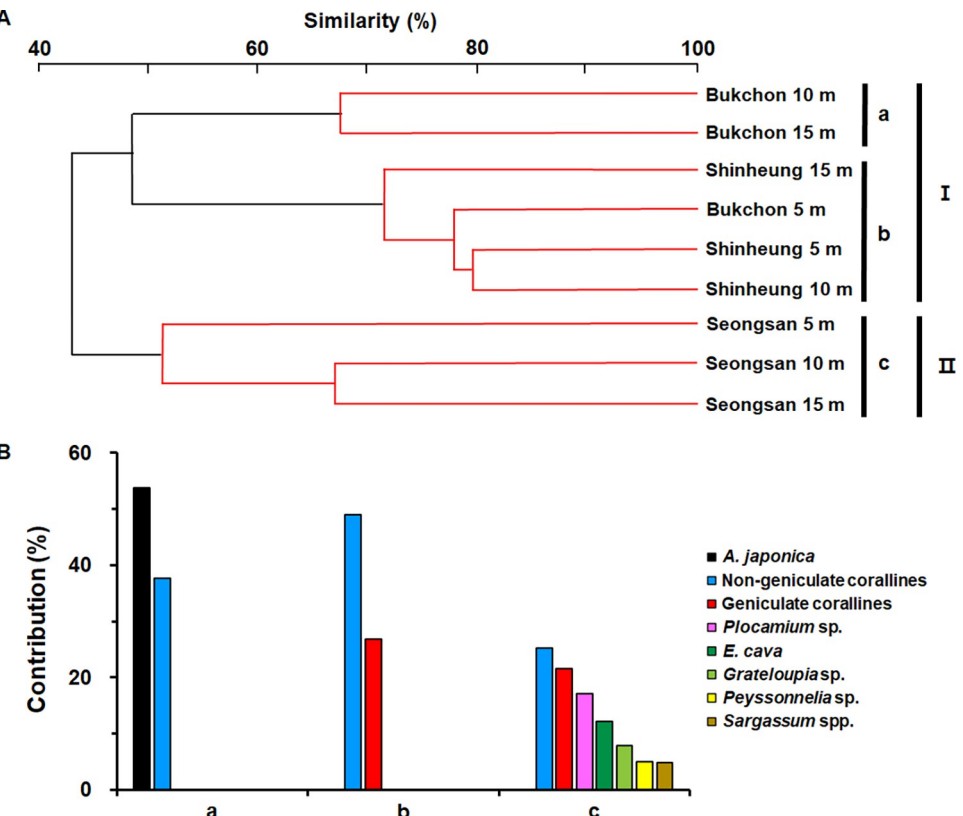

**Fig 7.** (A) Dendrogram of cluster analysis (using Bray and Curtis similarity) based on the percent cover of benthic organisms at depths of 5 m, 10 m, and 15 m of 3 survey sites SH, BC, and SS. Two major groups are detected (Group I and II). Red lines are indicated subgroups by the SIMPROF test. Three subgroups were detected (Subgroup a-c; $p < 0.005$). (B) The contributions of main benthic organisms (> 4%) among subgroups by SIMPER analysis.

subgroups (subgroup A-C; p<0.005) (Fig 7A). The three subgroups were characterized as follows: subgroup A including depths of 10 and 15 m at BC, subgroup B containing all depths at SH and 5m depth at BC, and subgroup C including all depths at SS. SIMPER revealed *that A. japonica* was a major species that contributed to the benthic community of subgroup A and non-geniculate corallines as a common component in subgroups A and B. In contrast, subgroup C consisted of various species such as geniculate corallines, *Plocamium* sp., *Ecklonia cava*, *Grateloupia* sp., *Peyssonnelia* sp., and *Sargassum* spp. including non-geniculate corallines (Fig 7B).

## Discussion

This study used high-resolution underwater images to identify and characterize benthic organisms living in the subtidal rocky substrate and extracted quantitative data (i.e., percent cover, density) on the benthic organisms, including *A. japonica*. Our analysis showed that the percent cover of *A. japonica* estimated at one of the sites (BC- at 15 m depth) was similar to those reported in the previous study [16]. The other 2 sites: Shinhueng and SS, were dominated by non-genticulates and macroalgae, respectively. The occurrence and dominance of *A. japonica* at some locations in Jeju is a recent phenomenon. Although few studies have hinted toward competition between *A. japonica* and macroalgae as the reason for its proliferation, this hypothesis is not yet proven. While it may be the case, other factors might have helped the

recruitment and proliferation of *A. japonica* in certain locations around Jeju Island. One of the factors might be the nutrient enrichment along the coast due to the presence of the land-based fish farms and the influence of the Tsushima current [47–53] and groundwater discharges [50, 54–58].

Along the coast of Jeju, the number of fish farms is more in the northern part of the island, see [53]; hence nutrient discharge may be more along the northern coast compared to that in the south or east. Previous studies that have reported an increased presence of *A. japonica* in Jeju are from locations with many fish farms (see locations shown in yellow in Fig 1; references). This may be one of the reasons for the abundance of *A. japonica* at BC and canopy-forming brown algae being rare at all depths surveyed (Figs 5 and 7). On the other hand, the abundance of *A. japonica* was low SS (5 to 10 m), while there was a high cover of macroalgal communities (canopy-forming brown algae, geniculate corallines, and other fleshy macroalgae) (Figs 5 and 7). So, rather than competition between *A. japonica* and kelp, it might be a selective presence of these two groups depending on the prevailing environmental conditions. As no studies have investigated the direct influence of nutrients (from groundwater discharges and fish farms) on the proliferation of *A. japonica* and how it affects the dynamics of species compositions of *A. japonica* and macroalgae, we can only guess that nutrient input into the sea in some locations (such as BC) in Jeju might be helping the increase in the abundance of *A. japonica*. Studies have shown that certain macroalgae persist due to high nutrients, while others cannot. High nutrient concentration can result in the bloom of species like *Ulva* spp., but at the same time result in low species number [59]. Due to fish farm effluent, macroalgal species in the intertidal zone are vulnerable, resulting in the decline of certain macroalgae (e.g., *Ulva* spp., *Eklonia cava*) at shallow depths–between 0–4 m) [59] this might also have been the case along the depth distribution gradient of *A. japonica* thereby creating space for recruitment and its subsequent spread. As studies have indicated, the nutrient enrichment does not seem to affect the populations of canopy-forming brown algae at deeper locations, and our observations also show that between 10–15 m depth, it is either the presence of *A. japonica* or kelp depending on location along the coastline of Jeju.

Another factor influencing changes in benthos dynamics is the increase in the average seawater temperatures, which might be responsible for the decrease in macroalgae population at several locations in Jeju Island [12, 16, 24, 60] and thus making way for the presence of *A. japonica* in the past couple of decades. In addition, increasing seawater temperature, see [11] might have affected the attachment and growth of new germlings. Serisawa et al. [2004] reported the decline in *E. cava* population in Tosa Bay (Kochi-Japan), associated with the rising seawater temperature [61], resulting in barren grounds. Hence, a combination of factors has resulted in the decline of kelp forests in the waters of Jeju Island, and non-geniculate corallines are rapidly occupying the newly available substrata [12, 16, 35, 62]. Denis et al. [2014] mentioned that crustose coralline algae (i.e., non-geniculate corallines) represent dominant taxa in areas not colonized by *A. japonica* in Jeju Island. In this study, non-geniculate corallines were the dominant taxa in the barren grounds, where macroalgae rarely appeared at BC and SH. Rocky substrates were found covered, predominantly by non-geniculate corallines, including various benthic fauna such as sponges and colonial ascidians, including *A. japonica* (See S1-S3 Figs in S2 File). We believe that over time, the loss of macroalgae might have resulted in the opportunistic settlement and proliferation of corals such as *A. japonica* due to high recruitment rates [12, 16].

Analysis of the photographs also showed rhodolith and crustose growth forms of non-geniculate corallines (Fig 2F). Both rhodolith and crustose growth forms have been reported to provide essential ecological functions towards the constitution of the benthic ecosystem [45]. Rhodolith forms provide an important rigid substrate for the attachment of benthic

macroalgae [63], and the growth of rhodolith is promoted by the shade from the canopy-forming kelp [64]. Kamenos et al. [2008] reported that the high latitude rhodolith has slow growth (200–300 μm year$^{-1}$). Temperate rhodolith are generally low light-adapted [45, 66], and exposure to high light reduces photosynthetic activity and subsequent bleaching response [45]. In contrast, crustose forms are essential components of tropical reef systems [67], providing suitable substrates or sufficient structures for coral larval settlement [45, 66, 67]. Tropical CCA are known to rapidly grow even under high light through shifts in their photoinhibition strategies according to light levels [44, 65–68]. Kim et al. [2011] documented that the cover of CCA, which were partly distributed in Jeju Island, increased 10.9% from 1998 to 2003 due to its persistence and continued northward range expansion.

Previous studies have documented abundance of *A. japonica* [11–13], reproduction [14], its recruitment [16], and its association with mollusks [17]. However, diversity of benthos, the abundance and relationship of *A. japonica* with the benthos has not been looked at. Although this study analyzed the benthos from a single time point (January 2014) using single transect at 3 depths and 3 locations, thereby constraining the resolution through space and time, the results of this study show differential presence of benthic communities depending on the sites and depth, more studies need to be conducted to understand the changes in the benthic community compositions through space and time with more replications, and long-term monitoring will be helpful in predicting the changes by documenting the expansion or contraction of benthic organisms.

The one difference in the data in this study compared with that of the previous studies [12, 16, 17] was differences in the density of *A. japonica*. These differences were probably due to several factors such as survey method, temporal differences, the area surveyed, and depth. [16] reported juvenile coral density by counting the single polyps observed at 15 m depth, representing a mean number of 7590 recruits m$^{-2}$. In this study, the minimum size of *A. japonica* colonies that could be observed and analyzed was generally equal to or more than 1 cm (See Fig 3), and it seems likely that the resolution of the camera used *in-situ* was insufficient to recognize the colonies (or polyps) that might have been smaller than 1cm. Diversity and distribution studies come with a caveat, and we acknowledge here that, on some occasions, camera image resolution cannot perform as expected due to uncontrollable underwater conditions. Also, the resolution of the underwater images obtained in this study was insufficient to distinguish geniculate from non-geniculate corallines in some cases. Thus, these organisms were classified as one functional group.

## Conclusion

This study utilized a non-destructive method to provide qualitative and quantitative information on the dynamics of benthic organisms at three sites in Jeju Island. The results showed that *A. japonica* dominates in the distribution, with high density at depths of 10 and 15 m of BC, where regionally endemic kelp forests almost disappeared and non-geniculate corallines covered the benthic substrate. Hence, in some locations, *A. japonica* and CCA opportunistically occupy the benthic substrates left vacant due to the decrease in the kelp forests in Jeju Island [12, 16]. Such dominance of new benthic groups might create an altered biogenic habitat potentially available for other benthic organisms by providing suitable substrates or sufficient structures [12, 17, 44]. Noseworthy et al. [2016] reported that *A. japonica* provides a habitat for twenty-seven bivalves and gastropods attached to the surface or incorporated inside the skeleton. This indicates that the benthic interactions are crucial to the diversity of the community and may be subject to change as the benthic community changes occur over time. Our results support the scenario of a recent *A. japonica* population increasing around Jeju Island from a

small previously existing population [12, 60]. The information obtained in this study contributes to our understanding of the major benthic community compositions in Jeju Island and provides a valuable baseline for exploring the interactions of benthic communities to future environmental changes.

## Supporting information

**S1 File.**
(PDF)

**S2 File.**
(DOCX)

## Acknowledgments

We thank the Shellfish Research and Aquaculture Laboratory members at Jeju National University for their assistance in the analysis. We also thank Mr. Sungwhan Cho of the Institute of Coastal Eco-Technology in Jeju, for SCUBA diving. We thank the 2 reviewers for their invaluable comments and suggestions, which helped improve the manuscript.

## Author Contributions

**Conceptualization:** Shashank Keshavmurthy, Kwang-Sik Choi.

**Data curation:** Hye-Mi Lee, Thatchaneshkanth Subramaniam, Hyun-Sung Yang, Sang Rul Park.

**Formal analysis:** Kyeong-Tae Lee, Hye-Mi Lee, Thatchaneshkanth Subramaniam, Hyun-Sung Yang, Sang Rul Park, Shashank Keshavmurthy.

**Funding acquisition:** Hyun-Sung Yang, Kwang-Sik Choi.

**Investigation:** Hye-Mi Lee, Shashank Keshavmurthy.

**Project administration:** Chang-Keun Kang, Kwang-Sik Choi.

**Resources:** Chang-Keun Kang, Kwang-Sik Choi.

**Software:** Kyeong-Tae Lee.

**Supervision:** Hyun-Sung Yang, Sang Rul Park, Chang-Keun Kang, Shashank Keshavmurthy, Kwang-Sik Choi.

**Visualization:** Shashank Keshavmurthy.

**Writing – original draft:** Kyeong-Tae Lee, Thatchaneshkanth Subramaniam, Hyun-Sung Yang, Sang Rul Park, Shashank Keshavmurthy, Kwang-Sik Choi.

**Writing – review & editing:** Kyeong-Tae Lee, Hyun-Sung Yang, Sang Rul Park, Chang-Keun Kang, Shashank Keshavmurthy, Kwang-Sik Choi.

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
