## [Decision Letter · Decision Letter 0]

31 Jan 2022

PONE-D-21-22570Dominance of the scleractinian coral Alveopora japonica in the barren subtidal hard bottom of high-latitude Jeju Island off the south coast of Korea assessed by high-resolution underwater imagesPLOS ONE

Dear Dr. Keshavmurthy,

Thank you for submitting your manuscript to PLOS ONE. After careful consideration, we feel that it has merit but does not fully meet PLOS ONE’s publication criteria as it currently stands. Therefore, we invite you to submit a revised version of the manuscript that addresses the points raised during the review process. Please accept my apologies for the time taken to provide you with reviews on your paper. It was extremely difficult to find reviewers under the current global situation. However, I am pleased that the two reviewers have provided very good reviews and extensive comments to improve the paper. Please pay particular attention to the suggestions about how to revise the discussion. There are several places where the discussion extends beyond the inference space of the study (e.g., inferring competition between corals and algae). Please revise throughout to remove such discussion and focus on the positive contribution of the data you present. Please submit your revised manuscript by Mar 17 2022 11:59PM. If you will need more time than this to complete your revisions, please reply to this message or contact the journal office at plosone@plos.org. Please include the following items when submitting your revised manuscript:A rebuttal letter that responds to each point raised by the academic editor and reviewer(s). You should upload this letter as a separate file labeled 'Response to Reviewers'.A marked-up copy of your manuscript that highlights changes made to the original version. You should upload this as a separate file labeled 'Revised Manuscript with Track Changes'.An unmarked version of your revised paper without tracked changes. You should upload this as a separate file labeled 'Manuscript'.

We look forward to receiving your revised manuscript.

Kind regards,

Bayden D. Russell

Academic Editor

PLOS ONE

Journal Requirements:

"We thank the Shellfish Research and Aquaculture Laboratory members at Jeju National University for their assistance in analysis. This study was supported by the “Long-term change of structure and function in the marine ecosystem of Korea” funded by the Ministry of Oceans and Fisheries of Korea. This research was also supported by Basic Science Research Program through the National Research Foundation of Korea (NRF) funded by the Ministry of Education (2019R1A6A1A03033553). We are also grateful to Mr. Sungwhan Cho of the Institute of Coastal Eco-Technology in Jeju, for SCUBA diving. SK is funded by the postdoctoral fellowship at Academia Sinica."

"KSC was partially funded by Basic Science Research Program through the National Research Foundation of Korea (NRF) funded by the Ministry of Education (2019R1A6A1A03033553). 

The funders had no role in study design, data collection and analysis, decision to publish, or preparation of the manuscript"

5. Please upload a copy of Figure 11, to which you refer in your text on page 10. If the figure is no longer to be included as part of the submission please remove all reference to it within the text.

6. We note that Figure 1 in your submission contain map images which may be copyrighted. All PLOS content is published under the Creative Commons Attribution License (CC BY 4.0), which means that the manuscript, images, and Supporting Information files will be freely available online, and any third party is permitted to access, download, copy, distribute, and use these materials in any way, even commercially, with proper attribution. For these reasons, we cannot publish previously copyrighted maps or satellite images created using proprietary data, such as Google software (Google Maps, Street View, and Earth). For more information, see our copyright guidelines: http://journals.plos.org/plosone/s/licenses-and-copyright.

Additional Editor Comments (if provided):

The manuscript presents some important and informative data on the expansion of corals into temperate regions concomitant with the contraction (loss) of macroalgae. However, the manuscript needs some major revision before it is publishable. Both reviewers provide substantial comments to guide the authors in these revisions. In particular, the authors need to ensure that they discuss the results within the inference space of surveys and do not infer processes like competition between the algal and coral communities.

Reviewers' comments:

Reviewer's Responses to Questions

**Comments to the Author**

1. Is the manuscript technically sound, and do the data support the conclusions?

Reviewer #1: Yes

Reviewer #2: Partly

2. Has the statistical analysis been performed appropriately and rigorously? 

Reviewer #1: Yes

Reviewer #2: Yes

3. Have the authors made all data underlying the findings in their manuscript fully available?

Reviewer #1: Yes

Reviewer #2: Yes

4. Is the manuscript presented in an intelligible fashion and written in standard English?

Reviewer #1: Yes

Reviewer #2: Yes

5. Review Comments to the Author

Reviewer #1: The authors present data on the distribution of the coral A. japonica and co-occurring macroalgae at three sites and three depth. The data is technically correct and of quality.

However the english should be checked as some sentence are grammatically incorrect or not clear. I have pointed out some of the mistakes (but not all).

The discussion is (and should be) simple as the paper mostly report on the distribution of the benthic algae and organisms. The main point of the discussion is that A. japonica occupy the space freed through the decrease in kelp and fucoids. The author also state that A. japonica can out compete macro-algae but not kelp and fucoids. However there is no proof of the out competition of other macroalgae by A. japonica. I think it is more that the recruitment (or growth) of A. japonica is inhibited by the presence of macroalgae (regardless if it is kelp or not) and perhaps promoted by the presence of cca. Perhaps the author do point it out a little in the discussion but it is really not clear.

Among the three sites studied, only at SH A. japonica showed the highest abundance at 5m. However I don't think this is discussed and could not find any specificity of this site to explain it, any idea? Is it the presence of the geniculate corallines?

## Intro:

page 9 - 1st paragraph: which is escalated by natural and anthropogenic activities. , It is not sure whether you are referring to climate change or the rearrangement of benthic community, please rephrase.

page 9 - 1st paragraph: instead of "altered species, which may appear for the first

time in ecosystems", what about "new interactions with species that are colonizing the ecosystems"

page 9: "According to [12]," should be changed by the Authors (Year)

page 9: "may result in the high density" indicate where these densities were observed. Same for the recruitment rates.

page 10: "has been reported by [17]." change to Author (year)

page 10: " A. japonica and coralline algae and kelp in Jeju" to " A. japonica and, coralline algae and kelp in Jeju" or "A. japonica, coralline algae and kelp in Jeju"

page 10: "While increased abundances could be a factor for kelp decline in Jeju" increased abundances of A. japonica?, please be clear. Also the two parts of this sentence are not linked. The first highlight the detrimental effect of A. japonica on kelp, and the other the fact that CCA increased. Was the CCA increase also detrimental to kelp as suggested by the word replaced?

page 10: "mass population" do you mean "important population"?

page 10: "However, through time, complex combinations of several causes such as rapid grazing by sea urchins, intense fishing or aquaculture activities, seawater pollutions, and global warming lead to a mass decline in the macroalgal population." Do you have a reference? I agree that these factors have for sure played a role, but without reference, it is better to rephrase to a more suggestive form. "however an important decline in the kelp population was observed in the last decades probably due to a complex combination of several causes such as rapid grazing by sea urchins, intense fishing or aquaculture activities, seawater pollutions, and global warming"

page 10: "In particular, growth rate metabolism, canopy-forming, and other physiological activities" to growth rate is not a thing, perhaps a comma is missing, but in this case what metabolism (photosynthesis, respiration, etc.) should be indicated. Also what in "canopy forming" do you mean the growth of the thalli and creation of the canopy?, this is not really a physiological activity in my opinion.

page 10: "the surveys due to high-resolution imagery" to "the surveys due to the availability of high-resolution imagery"

page 10: "ecology (29-32, 33) have shown " Cut the sentence before have shown, as now it is not grammatically correct. In what sense "data ouput is better", it contains more information, permanent record? give some example as it is can be quite subjective.

page 10: "is still manageable in their analysis in a short period" to "is still manageable and their analysis can be performed in a relatively short time"

page 11: "of different benthic organisms at three other sites and depths" remove other, it is not needed here.

*Note*: do author use photogrammetry?

## Mat & Met

page 12: "for provided additional light." to "to provide additional light"

Figure 3 caption: " The freehand regions were performed to the manual drawing of outline around A. japonica colonies" to "The freehand analysis consisted in the manual drawing of outline around A. japonica colonies"

page 14: "Coralline algae assemblages were identified conspicuous morphological characteristics based on their growth forms into non-geniculate growth form and geniculate growth form as described in (39)" to "Coralline algae assemblages were grouped based on their growth forms into non-geniculate growth form and geniculate growth form as described in (39)"

page 14: " 3) kelps (i.e., canopy-forming brown algae such as Ecklonia cava and Sargassum spp.)" Sargassum is not a kelp, it is a fucoid. Kelp are brown algae of the order Laminariales. Please rename your group to for example: canopy forming brown algae (includes kelp: E. cava and fucoids: Sargassum spp.)

## Results

page 14: "A total of 9 transects from three depths at three sites were filmed", I though photos were taken.. That is why you state in the method page 13: "outlined in 1m2 virtual quadrat". How did you get the scale from the video? Please indicate more clearly in the method that a video was taken, frame extracted and stitch together. Then virtual 1m2 quadrat were outline: Were these quadrats adjacent? How the scale (ie 1m) was determined?

page 15: "rhodophyte including Plocamium sp., Peyssonnelia sp chlorophyte including Cladophora sp. reported more than 5% average percentage cover in at least any of the sampling sites " I am slightly confused on whether you used the 6 groups (, 1) hard coral (exclusively A. japonica), 2) other sessile invertebrates , 3) kelps , 4) geniculate corallines algae, 5) non-geniculate corallines and 6) non-calcareous macroalgae) as highlighted in the method or the genus level data in your subsequent analysis. If you used the group, why not use the genus level?

page 16: "Contrary to BC, the coral density was 29.0 and 40.4 colonies m -2 at 5 m and 15 m depths at SH and SS, respectively" There should be 4 values here... or is it the mean of the two sites?

page 16: "whereas the highest percentage" should be "with the highest percentage"

Figure 7 caption : main benthic organisms (percentage coverage > 4%), now it is "<".

## Discussion

page 18: "the underwater images obtained in this study was insufficiently high enough to distinguish" to "the underwater images obtained in this study was insufficient to distinguish"

page 18: " that light is the most crucial factor driving the distribution" Do you mean the distribution of the brown morphs or the species itself? If the species, the morphs note is not required and only stating that "it is common for coral species to be found more abundantly at specific depth: ie Montipora capitata".

page 18-19: I am sorry but I don't see what information is discussed here. You didn't specifically check the association of the corals with Fugacium according to site or depth and so we do not know the physiological characteristics of this symbiont, so what advantage is provided? Better to remove as it is a completely open question that is not the topic of the paper.

page 20: " Especially, the rocky substrate were found covered, predominantly by non-geniculate corallines were observed including various benthic fauna such as sponges and colonial ascidians, including A. japonica (See supplementary Fig. 1-3)." This sentence is not grammatically correct and does not make sense at the moment, please rephrase.

page 20: "Alveopora japonica colonies were most abundant at BC, whereas canopy-forming kelps were rare at all depths." whereas should be change to "and". Whereas indicate an opposition between the two...

page 20: "successfully compete with benthic algae." Kelp is also a benthic algae, so your discussion on the growth form of A. japonica at the moment does not explain your statement: A. japonica can out-compete macroalgae except kelp.

page 20: " In addition, the massive and encrusting forms survive stressful conditions better than other coral forms from the community-structural shift in Japan" Not sure what you want to indicate here? What is the link with the current study?

page 20: "promoted by the shaded" should be "shade"

page 21: " (CCA) occupy a high proportion of the benthic bottom" in your study? They are easy to overgrow or they can easily overgrow other algae? Please rephrase.

page 21: "Tropical CCA are known fast-growing under high light through the ability of their dynamic photoinhibition strategies" to "Tropical CCA are known to be able to rapidly grow even under high light through their dynamic photoinhibition strategies" (<- may need to find something better as "dynamic photoinhibition strategies" is not really clear, perhaps "shift their photoinhibtion strategy according to light levels? )

Table 1: please indicate the site name in addition of the coordinate for your study so we can link the data presented in the table and in the text.

Reviewer #2: Overall, I believe the data are sound and present an interesting perspective of species turnover in a temperate marine habitat. However, I suggest that the discussion needs major revisions before publication. Several sections are difficult to understand, and the authors do not well link their data to the concepts presented. More importantly, the authors have not contextualized the differences in fauna across transects with differences in those environments specifically. There is an emphasis on competition, which was not looked at in the study, and not enough discussion of environmental parameters. For example, one study area is described as having lost macro-algae, turned barren, and then become dominated by coral. This means that coral and macroalgae never competed. It also implies a large role of the environmental change in excluding certain groups, while also likely becoming more habitable to corals. Among site comparisons should be developed and further described. BC and SH are across a bay from eachother, why are they so different? What is different about the environment in SS that supports kelp there and not in other places?

Minor points –

- Please add location names to table 1, also possibly add the other study locations to the map of Figure 1. The time component is interesting and mentioned in the discussion so these data should be clearly comparable.

- Would be interesting to see the bleaching/dead colonies visualized in Fig. 4 as well if possible.

- There seems to be some interchanging between Groups 1 & 2, subgroups A-C in the cluster analysis and groups I, II, III in Figure 7. Please revisit the nomenclature here.

- There is also some switching in how the sites are referred to, either as abbreviations or full names. I think the names could be used within the manuscript without difficulty for the reader.

“In this study, bleaching and dead colonies of A. japonica at shallow depths (i.e., depths of 5 and 10 m) in a high light environment were higher than that at a deeper depth. This may be the reason for the abundance and preference of A. japonica to thrive at depths of 15 m. (42) have shown that the coral Montipora capitata occurs as orange and brown color morphs due to the strong influence of depth and symbiont associations and that light is the most crucial factor driving the distribution.”

- I am really having a hard time understanding this paragraph. Both from the perspective of meaning and relevance. Meaning: I think it needs to be more clear where you are talking about depth, light, and abundance. Relevance: many corals have specific depth distributions, it’s unclear what the comparison to M. capitata is meant to imply. Lastly, I would not use the word “preference” here.

“Another factor that influences the success of a coral species is its association with Symbiodiniaceae, which confer competitive advantages to its host (43). In the case of A. japonica, depending on the location, they are known to associate with Cladocopium and Fugacium species, respectively (44-46), and in some cases, both Cladocopium sp. and Fugacium sp. simultaneously (44). Previous study (47) has shown that A. japonica in Jeju are associated with Fugacium sp. and although the physiological role of Fugacium sp. is still not resolved, we believe that its association with A. japonica in Jeju might be advantageous and hence contribute to specific activities depending on environmental conditions (44, 48). Future studies to understand the distribution of A. japonica among depths and a more detailed analysis of associated Symbiodiniaceae using high-resolution tools can provide better information on their dominance and increasing population in Jeju Island.“

- What is the competitive advantage referring to? Competitive relative to other benthic organisms? Competitive relative to A. japonica associating with Cladocopium? I don’t understand the implication here. Also, Palmas et al. (2015) did a relatively exhaustive survey (78 colonies across 8 sites) and genotyping of symbionts in A. japonica in Jeju, what is the expectation of “more detailed analysis of associated Symbiodiniaceae”?

“A comparison of the benthic communities between BC and SS indicated that the space competition was observed for A. japonica, which interacted with kelps and rarely won, whereas competed successfully against other macroalgae, resulting in a higher relative abundance of A. japonica at BC.”

- The authors did not study competition, it’s unclear to me how they are reporting on this and who is “winning” from a snapshot transect.

“Alveopora japonica growth forms are sub-massive, massive, or hemispherical (57). According to (58), coral growth forms associated with slow growth (i.e., massive, encrusting growth form) successfully compete with benthic algae. In addition, the massive and encrusting forms survive stressful conditions better than other coral forms from the community-structural shift in Japan (59).“

- What are the stressful conditions here? Environmental or competitive? Competitive with other corals? Japan is an interesting comparison but there needs to be more context here, what other corals? Are there branching corals in Jeju? Is their abundance increasing also?

“These changes could have a major irreversible impact on the benthic ecosystem in Jeju Island.”

- Like what? The authors could be more explicit about whether those CCA’s can promote coral recruitment vs. kelp recruitment, for example.

- In the discussion I would present first, your data and context of how your sites/depths of your surveys compare to eachother. Comparisons out to other literature, and limitations of the methods in making those comparisons can come later.

- The paper would benefit from minor English editing. Unfortunately I see that this is not something that the journal will assist with. A careful read through from authors may be enough. Most of the time it did not affect my ability to understand the authors’ meaning but in a few places in the intro and particularly the discussion as outlined above, it may cause some misinterpretations, as outlined below.

6. PLOS authors have the option to publish the peer review history of their article (what does this mean?). If published, this will include your full peer review and any attached files.

Reviewer #1: **Yes: **Sylvain Agostini

Reviewer #2: **Yes: **Shelby McIlroy

---

## [Author Response · Author response to Decision Letter 0]

4 Aug 2022

Reviewer #1: 

The authors present data on the distribution of the coral A. japonica and co-occurring macroalgae at three sites and three depth. The data is technically correct and of quality.

However the english should be checked as some sentence are grammatically incorrect or not clear. I have pointed out some of the mistakes (but not all).

The discussion is (and should be) simple as the paper mostly report on the distribution of the benthic algae and organisms. The main point of the discussion is that A. japonica occupy the space freed through the decrease in kelp and fucoids. The author also state that A. japonica can out compete macro-algae but not kelp and fucoids. However there is no proof of the out competition of other macroalgae by A. japonica. I think it is more that the recruitment (or growth) of A. japonica is inhibited by the presence of macroalgae (regardless if it is kelp or not) and perhaps promoted by the presence of cca. Perhaps the author do point it out a little in the discussion but it is really not clear.

Among the three sites studied, only at SH A. japonica showed the highest abundance at 5m. However I don't think this is discussed and could not find any specificity of this site to explain it, any idea? Is it the presence of the geniculate corallines?

Authors: We thank the reviewer for their comments and suggestions. For the revised submission, we have got the manuscript edited by an English reviewer. Also, we have revised the discussion as per the suggestions. We agree that it is not suitable to mention competition as we have not done studies related to competition. Hence, this aspect has been removed from the text.

## Intro:

page 9 - 1st paragraph: which is escalated by natural and anthropogenic activities. , It is not sure whether you are referring to climate change or the rearrangement of benthic community, please rephrase.

Authors: We have rephrased this sentence.

page 9 - 1st paragraph: instead of "altered species, which may appear for the first

time in ecosystems", what about "new interactions with species that are colonizing the ecosystems"

Authors: Thank you for this suggestion. The sentence has been revised

page 9: "According to [12]," should be changed by the Authors (Year)

Authors: Thank you for this suggestion. The sentence has been revised

page 9: "may result in the high density" indicate where these densities were observed. Same for the recruitment rates.

Authors: Thank you for this comment. The information has been added.

page 10: "has been reported by [17]." change to Author (year)

Authors: We have changed the sentence to keep the references in the bracket at per PLOS one requirements

page 10: " A. japonica and coralline algae and kelp in Jeju" to " A. japonica and, coralline algae and kelp in Jeju" or "A. japonica, coralline algae and kelp in Jeju"

Authors: : Thank you for this suggestion. The sentence has been revised

page 10: "While increased abundances could be a factor for kelp decline in Jeju" increased abundances of A. japonica?, please be clear. Also the two parts of this sentence are not linked. The first highlight the detrimental effect of A. japonica on kelp, and the other the fact that CCA increased. Was the CCA increase also detrimental to kelp as suggested by the word replaced?

Authors: : Thank you for this suggestion. The sentence has been revised

page 10: "mass population" do you mean "important population"?

Authors: : Thank you for this suggestion. The sentence has been revised

page 10: "However, through time, complex combinations of several causes such as rapid grazing by sea urchins, intense fishing or aquaculture activities, seawater pollutions, and global warming lead to a mass decline in the macroalgal population." Do you have a reference? I agree that these factors have for sure played a role, but without reference, it is better to rephrase to a more suggestive form. "however an important decline in the kelp population was observed in the last decades probably due to a complex combination of several causes such as rapid grazing by sea urchins, intense fishing or aquaculture activities, seawater pollutions, and global warming"

Authors: : Thank you for this suggestion. The sentence has been revised

page 10: "In particular, growth rate metabolism, canopy-forming, and other physiological activities" to growth rate is not a thing, perhaps a comma is missing, but in this case what metabolism (photosynthesis, respiration, etc.) should be indicated. Also what in "canopy forming" do you mean the growth of the thalli and creation of the canopy?, this is not really a physiological activity in my opinion.

Authors: : Thank you for this suggestion. The sentence has been revised

page 10: "the surveys due to high-resolution imagery" to "the surveys due to the availability of high-resolution imagery"

Authors: : Thank you for this suggestion. The sentence has been revised

page 10: "ecology (29-32, 33) have shown " Cut the sentence before have shown, as now it is not grammatically correct. In what sense "data ouput is better", it contains more information, permanent record? give some example as it is can be quite subjective.

Authors: : Thank you for this suggestion. The sentence has been revised

page 10: "is still manageable in their analysis in a short period" to "is still manageable and their analysis can be performed in a relatively short time"

Authors: : Thank you for this suggestion. The sentence has been revised

page 11: "of different benthic organisms at three other sites and depths" remove other, it is not needed here.

Authors: : Thank you for this suggestion. The sentence has been revised

*Note*: do author use photogrammetry?

Authors: : Good point. We did not use photogrammetry. The word has been removed from the text

## Mat & Met

page 12: "for provided additional light." to "to provide additional light"

Authors: : Thank you for this suggestion. The sentence has been revised

Figure 3 caption: " The freehand regions were performed to the manual drawing of outline around A. japonica colonies" to "The freehand analysis consisted in the manual drawing of outline around A. japonica colonies"

Authors: : Thank you for this suggestion. The sentence has been revised

page 14: "Coralline algae assemblages were identified conspicuous morphological characteristics based on their growth forms into non-geniculate growth form and geniculate growth form as described in (39)" to "Coralline algae assemblages were grouped based on their growth forms into non-geniculate growth form and geniculate growth form as described in (39)"

Authors: : Thank you for this suggestion. The sentence has been revised

page 14: " 3) kelps (i.e., canopy-forming brown algae such as Ecklonia cava and Sargassum spp.)" Sargassum is not a kelp, it is a fucoid. Kelp are brown algae of the order Laminariales. Please rename your group to for example: canopy forming brown algae (includes kelp: E. cava and fucoids: Sargassum spp.)

Authors: : Thank you for this suggestion. The sentence has been revised

## Results

page 14: "A total of 9 transects from three depths at three sites were filmed", I though photos were taken.. That is why you state in the method page 13: "outlined in 1m2 virtual quadrat". How did you get the scale from the video? Please indicate more clearly in the method that a video was taken, frame extracted and stitch together. Then virtual 1m2 quadrat were outline: Were these quadrats adjacent? How the scale (ie 1m) was determined?

Authors – We did not conduct video transects. Actually, 20 high-resolution images were captured (1m x1m) continuously over a 20ｘ1 m line transect installed at 1 m intervals during every 30 minutes dives using SCUBA with a digital underwater camera, at depths of 5, 10, 15 m, respectively. This part of the explanation has been revised. We also have included “information on image processing” as a supplementary file.

page 15: "rhodophyte including Plocamium sp., Peyssonnelia sp chlorophyte including Cladophora sp. reported more than 5% average percentage cover in at least any of the sampling sites " I am slightly confused on whether you used the 6 groups (, 1) hard coral (exclusively A. japonica), 2) other sessile invertebrates , 3) kelps , 4) geniculate corallines algae, 5) non-geniculate corallines and 6) non-calcareous macroalgae) as highlighted in the method or the genus level data in your subsequent analysis. If you used the group, why not use the genus level?

Authors: Thank you for bringing this point to our attention. After revisiting this whole part did not make sense and hence, we have revised this part. Also we have now replaced “Kelp” with “canopy-forming brown algae” and “non-calcareous macroalgae” with “Oher fleshy macroalgae”

page 16: "Contrary to BC, the coral density was 29.0 and 40.4 colonies m -2 at 5 m and 15 m depths at SH and SS, respectively" There should be 4 values here... or is it the mean of the two sites?

Authors: : Sorry for the confusion. The sentence has been revised

page 16: "whereas the highest percentage" should be "with the highest percentage"

Authors: : Thank you. The sentence has been revised

Figure 7 caption : main benthic organisms (percentage coverage > 4%), now it is "<".

Authors: : Sorry for the mistake. It is now corrected

## Discussion

page 18: "the underwater images obtained in this study was insufficiently high enough to distinguish" to "the underwater images obtained in this study was insufficient to distinguish"

Authors: : Sorry for the confusion. The sentence has been revised

page 18: " that light is the most crucial factor driving the distribution" Do you mean the distribution of the brown morphs or the species itself? If the species, the morphs note is not required and only stating that "it is common for coral species to be found more abundantly at specific depth: ie Montipora capitata".

Authors: : We agree with the reviewer. This information was confusing and not necessary. We have removed it from the text

page 18-19: I am sorry but I don't see what information is discussed here. You didn't specifically check the association of the corals with Fugacium according to site or depth and so we do not know the physiological characteristics of this symbiont, so what advantage is provided? Better to remove as it is a completely open question that is not the topic of the paper.

Authors: : We agree with the reviewer. This information was not necessary. We have removed it from the text 

page 20: " Especially, the rocky substrate were found covered, predominantly by non-geniculate corallines were observed including various benthic fauna such as sponges and colonial ascidians, including A. japonica (See supplementary Fig. 1-3)." This sentence is not grammatically correct and does not make sense at the moment, please rephrase.

Authors: : Thank you for this suggestion. The sentence has been revised

page 20: "Alveopora japonica colonies were most abundant at BC, whereas canopy-forming kelps were rare at all depths." whereas should be change to "and". Whereas indicate an opposition between the two...

Authors: : Thank you for this suggestion. The sentence has been revised

page 20: "successfully compete with benthic algae." Kelp is also a benthic algae, so your discussion on the growth form of A. japonica at the moment does not explain your statement: A. japonica can out-compete macroalgae except kelp.

Authors: : Thank you for this suggestion. This part of the discussion has been revised.

page 20: " In addition, the massive and encrusting forms survive stressful conditions better than other coral forms from the community-structural shift in Japan" Not sure what you want to indicate here? What is the link with the current study?

Authors: : Thank you for pointing this out. We have removed it from the text, it was not necessary to use it

page 20: "promoted by the shaded" should be "shade"

Authors: : Thank you for this suggestion. The sentence has been revised

page 21: " (CCA) occupy a high proportion of the benthic bottom" in your study? They are easy to overgrow or they can easily overgrow other algae? Please rephrase.

page 21: "Tropical CCA are known fast-growing under high light through the ability of their dynamic photoinhibition strategies" to "Tropical CCA are known to be able to rapidly grow even under high light through their dynamic photoinhibition strategies" (<- may need to find something better as "dynamic photoinhibition strategies" is not really clear, perhaps "shift their photoinhibtion strategy according to light levels? )

Authors: : Thank you for this suggestion. The sentence has been revised

Table 1: please indicate the site name in addition of the coordinate for your study so we can link the data presented in the table and in the text.

Authors: : Thank you for this suggestion. The Table 1 has been revised

Reviewer #2: 

Overall, I believe the data are sound and present an interesting perspective of species turnover in a temperate marine habitat. However, I suggest that the discussion needs major revisions before publication. Several sections are difficult to understand, and the authors do not well link their data to the concepts presented. More importantly, the authors have not contextualized the differences in fauna across transects with differences in those environments specifically. There is an emphasis on competition, which was not looked at in the study, and not enough discussion of environmental parameters. For example, one study area is described as having lost macro-algae, turned barren, and then become dominated by coral. This means that coral and macroalgae never competed. It also implies a large role of the environmental change in excluding certain groups, while also likely becoming more habitable to corals. Among site comparisons should be developed and further described. BC and SH are across a bay from eachother, why are they so different? What is different about the environment in SS that supports kelp there and not in other places?

Authors: We thank the reviewer for their comments and suggestions. For the revised submission, we have got the manuscript edited by an English reviewer. Also, we have revised the discussion as per the suggestions. We agree that it is not suitable to mention competition as we have not done studies related to competition. Hence, this aspect has been removed from the text.

Minor points –

- Please add location names to table 1, also possibly add the other study locations to the map of Figure 1. The time component is interesting and mentioned in the discussion so these data should be clearly comparable.

Authors: : Thank you for this suggestion. Table 1 has been revised. We did not add the locations in the previous study as all those studies have map showing the locations. Also, including them may confuse some readers. We would like to stick with the present figure.

- Would be interesting to see the bleaching/dead colonies visualized in Fig. 4 as well if possible.

Authors: : Thank you, this is a good suggestion. We have revised Figure 4 to included bleached/dead colonies

- There seems to be some interchanging between Groups 1 & 2, subgroups A-C in the cluster analysis and groups I, II, III in Figure 7. Please revisit the nomenclature here.

Authors: : Thank you for bringing this issue to our attention. We have revised Figure 7

- There is also some switching in how the sites are referred to, either as abbreviations or full names. I think the names could be used within the manuscript without difficulty for the reader.

Authors: : Thank you for this suggestion. This is good point. We now use full name of the sites everywhere in the text, figures and table

“In this study, bleaching and dead colonies of A. japonica at shallow depths (i.e., depths of 5 and 10 m) in a high light environment were higher than that at a deeper depth. This may be the reason for the abundance and preference of A. japonica to thrive at depths of 15 m. (42) have shown that the coral Montipora capitata occurs as orange and brown color morphs due to the strong influence of depth and symbiont associations and that light is the most crucial factor driving the distribution.”

- I am really having a hard time understanding this paragraph. Both from the perspective of meaning and relevance. Meaning: I think it needs to be more clear where you are talking about depth, light, and abundance. Relevance: many corals have specific depth distributions, it’s unclear what the comparison to M. capitata is meant to imply. Lastly, I would not use the word “preference” here.

Authors: : We agree with the reviewer. Some part of the discussion was confusing, and some information were not necessary. We have revised the discussion as per the suggestions from both the reviewers.

“Another factor that influences the success of a coral species is its association with Symbiodiniaceae, which confer competitive advantages to its host (43). In the case of A. japonica, depending on the location, they are known to associate with Cladocopium and Fugacium species, respectively (44-46), and in some cases, both Cladocopium sp. and Fugacium sp. simultaneously (44). Previous study (47) has shown that A. japonica in Jeju are associated with Fugacium sp. and although the physiological role of Fugacium sp. is still not resolved, we believe that its association with A. japonica in Jeju might be advantageous and hence contribute to specific activities depending on environmental conditions (44, 48). Future studies to understand the distribution of A. japonica among depths and a more detailed analysis of associated Symbiodiniaceae using high-resolution tools can provide better information on their dominance and increasing population in Jeju Island.“

- What is the competitive advantage referring to? Competitive relative to other benthic organisms? Competitive relative to A. japonica associating with Cladocopium? I don’t understand the implication here. Also, Palmas et al. (2015) did a relatively exhaustive survey (78 colonies across 8 sites) and genotyping of symbionts in A. japonica in Jeju, what is the expectation of “more detailed analysis of associated Symbiodiniaceae”?

Authors: : We agree with the reviewer. This information was not necessary. We have revised the discussion as per the suggestions from both the reviewers.

“A comparison of the benthic communities between BC and SS indicated that the space competition was observed for A. japonica, which interacted with kelps and rarely won, whereas competed successfully against other macroalgae, resulting in a higher relative abundance of A. japonica at BC.”

- The authors did not study competition, it’s unclear to me how they are reporting on this and who is “winning” from a snapshot transect.

Authors: : Yes, as stated in the beginning, we have not looked at competition and hence it is not relevant to discuss it. We agree with the reviewer. We have revised the discussion as per the suggestions from both the reviewers.

“Alveopora japonica growth forms are sub-massive, massive, or hemispherical (57). According to (58), coral growth forms associated with slow growth (i.e., massive, encrusting growth form) successfully compete with benthic algae. In addition, the massive and encrusting forms survive stressful conditions better than other coral forms from the community-structural shift in Japan (59).“

- What are the stressful conditions here? Environmental or competitive? Competitive with other corals? Japan is an interesting comparison but there needs to be more context here, what other corals? Are there branching corals in Jeju? Is their abundance increasing also?

Authors: : This part of the discussion has been revised by removing irrelevant information with respect to this study.

“These changes could have a major irreversible impact on the benthic ecosystem in Jeju Island.”

- Like what? The authors could be more explicit about whether those CCA’s can promote coral recruitment vs. kelp recruitment, for example.

Authors: : Thank you for the suggestion. We have revised the discussion.

- In the discussion I would present first, your data and context of how your sites/depths of your surveys compare to eachother. Comparisons out to other literature, and limitations of the methods in making those comparisons can come later.

Authors: : Thank you for the suggestion. We have revised the discussion.

- The paper would benefit from minor English editing. Unfortunately I see that this is not something that the journal will assist with. A careful read through from authors may be enough. Most of the time it did not affect my ability to understand the authors’ meaning but in a few places in the intro and particularly the discussion as outlined above, it may cause some misinterpretations, as outlined below.

---

## [Editor Report · Decision Letter 1]

6 Sep 2022

PONE-D-21-22570R1Dominance of the scleractinian coral Alveopora japonica in the barren subtidal hard bottom of high-latitude Jeju Island off the south coast of Korea assessed by high-resolution underwater imagesPLOS ONE

Dear Dr. Keshavmurthy,

Thank you for submitting your manuscript to PLOS ONE. After careful consideration, we feel that it has merit but does not fully meet PLOS ONE’s publication criteria as it currently stands. Therefore, we invite you to submit a revised version of the manuscript that addresses the points raised during the review process. Thank you for the substantial revisions that you have made on your manuscript; it is much improved. There are still some minor issues that need to be fixed before the ms is acceptable for publication (see the attached comments from the editor).

We look forward to receiving your revised manuscript.

Kind regards,

Bayden D. Russell

Academic Editor

PLOS ONE

Journal Requirements:

Additional Editor Comments:

Thank you for the substantial revisions of your manuscript. It is substantially improved.

There are still some small issues that need to be fixed before the paper is acceptable for publication:

1. The abstract needs a motivating sentence or two to start which gives some background and context for the study. The new revised sentence that starts the abstract jumps straight into what sounds like results without any context. I found that the sentence(s) that started the abstract in the original version of the paper were better for this.

2. Methods. It is unclear in the methods whether there was a single 1 x 20 m photo transect per depth and site. Please clarify in the methods whether there were any replicate transects.

If not, then there needs to be explicit discussion (in the discussion section) on the lack of spatial and seasonal replication of your surveys and what this may mean for your analysis of the benthic communities.

3. Methods and Results. Please revise the methods and results sections to ensure that there is always a space between the "times" symbol and numbers, and numbers and the units. There are many instances, for example, of "1 x1m". There should be spaces between the numbers and symbols/units. Also, it looks like you have used the small letter "x" for the "times" symbol. You should use the actual symbol (found in the symbols in the "insert" menu on ms Word).

---

## [Author Response · Author response to Decision Letter 1]

8 Sep 2022

According to your comments and suggestions, we have revised parts of the manuscript with following changes and additions.

1. Abstract – We have included the opening sentence with a context

2. Methods – About the clarity of use of transects – we have not made it clear in the methods that a single transect was used at a single time point. In the discussion we have included

“Previous studies have documented abundance of A. japonica (11-13), reproduction (14), its recruitment (16), and its association with mollusks (17). However, diversity of benthos, the abundance and relationship of A. japonica with the benthos has not been looked at. Although this study analyzed the benthos from a single time point (January 2014) using single transect at 3 depths and 3 locations, thereby constraining the resolution through space and time, the results of this study show differential presence of benthic communities depending on the sites and depth, more studies need to be conducted to understand the changes in the benthic community compositions through space and time with more replications, and long-term monitoring will be helpful in predicting the changes by documenting the expansion or contraction of benthic organisms”

3. With respect to “times” symbol, we have not replaced it with proper symbol with space in between

---

## [Editor Report · Decision Letter 2]

13 Sep 2022

Dominance of the scleractinian coral Alveopora japonica in the barren subtidal hard bottom of high-latitude Jeju Island off the south coast of Korea assessed by high-resolution underwater images

PONE-D-21-22570R2

Dear Dr. Keshavmurthy,

We’re pleased to inform you that your manuscript has been judged scientifically suitable for publication and will be formally accepted for publication once it meets all outstanding technical requirements.

Kind regards,

Bayden D. Russell

Academic Editor

PLOS ONE
---

## [Editor Report · Acceptance letter]

25 Oct 2022

PONE-D-21-22570R2 

Dominance of the scleractinian coral *Alveopora japonica* in the barren subtidal hard bottom of high-latitude Jeju Island off the south coast of Korea assessed by high-resolution underwater images 

Dear Dr. Keshavmurthy:

I'm pleased to inform you that your manuscript has been deemed suitable for publication in PLOS ONE. Congratulations! Your manuscript is now with our production department. 

Kind regards, 

on behalf of

Dr. Bayden D. Russell 

Academic Editor

PLOS ONE